# Sample Selection with Uncertainty of Losses for Learning with Noisy Labels

## Abstract

In learning with noisy labels, the *sample selection* approach is very popular, which regards *small-loss* data as correctly labeled during training. However, losses are generated on-the-fly based on the model being trained with noisy labels, and thus *large-loss* data are *likely but not certainly* to be incorrect. There are actually two possibilities of a large-loss data point: (a) it is mislabeled, and then its loss *decreases slower* than other data, since deep neural networks "learn patterns first"; (b) it belongs to an underrepresented group of data and *has not been selected yet*. In this paper, we incorporate the uncertainty of losses by adopting *interval estimation* instead of *point estimation* of losses, where lower bounds of the *confidence intervals* of losses derived from *distribution-free concentration inequalities*, but not losses themselves, are used for sample selection. In this way, we also give large-loss but less selected data a try; then, we can better distinguish between the cases (a) and (b) by seeing if the losses *effectively decrease* with the uncertainty after the try. As a result, we can better explore underrepresented data that are correctly labeled but seem to be mislabeled *at first glance*. Experiments demonstrate that the proposed method is superior to baselines and robust to a broad range of label noise types.

## 1 Introduction

Learning with noisy labels is one of the most challenging problems in weakly-supervised learning, since noisy labels are ubiquitous in the real world [36, 65, 40, 1, 61]. For instance, both crowdsourcing and web crawling yield large numbers of noisy labels everyday [12]. Noisy labels can severely impair the performance of deep neural networks with strong memorization capacities [67, 69, 42, 30].

To reduce the influence of noisy labels, a lot of approaches have been recently proposed [38, 29, 31, 68, 71, 55, 56, 46, 33, 25, 34, 47, 60, 49, 19, 17, 14]. They can be generally divided into two main categories. The first one is to estimate the noise transition matrix [41, 44, 15, 11], which denotes the probabilities that clean labels flip into noisy labels. However, the noise transition matrix is hard to be estimated accurately, especially when the number of classes is large [65]. The second approach is sample selection, which is *our focus* in this paper. This approach is based on selecting possibly clean examples from a mini-batch for training [12, 62, 50, 65, 23, 50, 51]. Intuitively, if we can exploit less noisy data for network parameter updates, the network will be more robust.

A major question in sample selection is what *criteria* can be used to select possibly clean examples. At the present stage, the selection based on the *small-loss* criteria is the most common method, and has been verified to be effective in many circumstances [12, 16, 65, 52, 62]. Specifically, since deep networks *learn patterns first* [2], they would first memorize training data of clean labels and then those of noisy labels with the assumption that clean labels are of the majority in a noisy class. Small-loss examples can thus be regarded as clean examples *with high probability*. Therefore, in each iteration, prior methods [12, 52] select the small-loss examples based on *the predictions of the current network* for robust training.

Submitted to 35th Conference on Neural Information Processing Systems (NeurIPS 2021). Do not distribute.

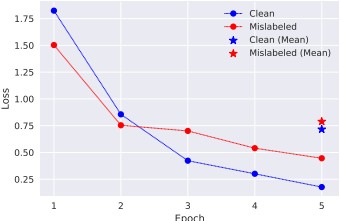 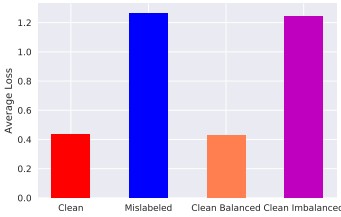

Figure 1: Illustrations of *uncertainty of losses*. Experiments are conducted on the imbalanced noisy *MNIST* dataset. **Left**: uncertainty of *small-loss* examples. At the beginning of training (Epochs 1 and 2), due to the instability of the current prediction, the network gives a larger loss to the clean example and does not select it for updates. If we consider the *mean* of training losses at different epochs, the clean example can be equipped with a smaller loss and then selected for updates. **Right**: uncertainty of *large-loss* examples. Since the deep network learns easy examples at the beginning of training, it gives a large loss to *clean imbalanced* data with non-dominant labels, which causes such data unable to be selected and severely influence generalization.

However, such a selection procedure is *debatable*, since it arguably does *not consider uncertainty* in selection. The uncertainty comes from two aspects. First, this procedure has *uncertainty about small-loss examples*. Specifically, the procedure uses *limited time intervals* and only exploits the losses provided by the *current predictions*. For this reason, the estimation for the noisy class posterior is *unstable* [63], which causes the network predictions to be equally unstable. It thus *takes huge risks* to only use losses provided by the current predictions (Figure 1, left). Once wrong selection is made, the inferiority of accumulated errors will arise [65]. Second, this procedure has *uncertainty about large-loss examples*. To be specific, deep networks learn easy examples at the beginning of training, but ignore some clean examples with large losses. Nevertheless, such examples are always critical for generalization. For instance, when learning with *imbalanced* data, distinguishing the examples with *non-dominant labels* are more pivotal during training [35]. Deep networks often give large losses to such examples (Figure 1, right). Therefore, when learning under the realistic scenes, e.g., learning with noisy imbalanced data, prior sample selection methods cannot address such an issue well.

To relieve the above issues, we study the uncertainty of losses in the sample selection procedure to combat noisy labels. To reduce the uncertainty of small-loss examples, we extend time intervals and utilize the *mean* of training losses at different training iterations. In consideration of the bad influence of mislabeled data on training losses, we build two *robust mean estimators* from the perspectives of *soft truncation* and *hard truncation* w.r.t. the truncation level, respectively. Soft truncation makes the mean estimation more robust by *holistically* changing the behavior of losses. Hard truncation makes the mean estimation more robust by *locally* removing outliers from losses. To reduce the uncertainty of large-loss examples, we encourage networks to pick the sample that has not been selected in a conservative way. Furthermore, to address the two issues *simultaneously*, we derive *concentration inequalities* [5] for robust mean estimation and further employ statistical *confidence bounds* [3] to consider the number of times an example was selected during training.

The study of uncertainty of losses in learning with noisy labels can be justified as follows. In statistical learning, it is known that uncertainty is related to the quality of data [48]. Philosophically, we need *variety decrease* for selected data and *variety search* for unselected data, which share a common objective, i.e., *reduce the uncertainty of data to improve generalization* [37]. This is our original intention, since noisy labels could bring more uncertainty because of the low quality of noisy data. Nevertheless, due to the harm of noisy labels for generalization, we need to strike a good balance between variety decrease and search. Technically, our method is specially designed for handling noisy labels, which robustly uses network predictions and conservatively seeks less selected examples meanwhile to reduce the uncertainty of losses and then generalize well.

Before delving into details, we clearly emphasize our contributions in two folds. First, we reveal prior sample selection criteria in learning with noisy labels have some potential weaknesses and discuss them in detail. The new selection criteria are then proposed with detailed theoretical analyses. Second, we experimentally validate the proposed method on both synthetic noisy balanced/imbalanced datasets and real-world noisy datasets, on which it achieves superior robustness compared with the state-of-the-art methods in learning with noisy labels. The rest of the paper is organized as follows. In Section 2, we propose our robust learning paradigm step by step. Experimental results are discussed in Section 3. The conclusion is given in Section 4.

## 2 Method

In this section, we first introduce the problem setting and some background (Section 2.1). Then we discuss how to exploit training losses at different iterations (Section 2.2). Finally, we introduce the proposed method, which exploits training losses at different iterations more robustly and encourages networks to pick the sample that is less selected but could be correctly labeled (Section 2.3).

### 2.1 Preliminaries

Let $\mathcal{X}$ and $\mathcal{Y}$ be the input and output spaces. Consider a $k$-class classification problem, i.e., $\mathcal{Y} = [k]$, where $[k] = \{1, \ldots, k\}$. In learning with noisy labels, the training data are all sampled from a corrupted distribution on $\mathcal{X} \times \mathcal{Y}$. We are given a sample with noisy labels, i.e., $\tilde{S} = \{(\mathbf{x}, \tilde{y})\}$, where $\tilde{y}$ is the noisy label. The aim is to learn a robust classifier that could assign clean labels to test data by only exploiting a training sample with noisy labels.

Let $f : \mathcal{X} \to \mathbb{R}^k$ be the classifier with learnable parameters $\mathbf{w}$. At the $i$-th iteration during training, the parameters of the classifier $f$ can be denoted as $\mathbf{w}_i$. Let $\ell : \mathbb{R}^k \times \mathcal{Y} \to \mathbb{R}$ be a *surrogate loss function* for $k$-class classification. We exploit the *softmax cross entropy loss* in this paper. Given an arbitrary training example $(\mathbf{x}, \tilde{y})$, at the $i$-th iteration, we can obtain a loss $\ell_i$, i.e., $\ell_i = \ell(f(\mathbf{w}_i; \mathbf{x}), \tilde{y})$. Hence, until the $t$-th iteration, we can obtain a training loss set $L_t$ about the example $(\mathbf{x}, \tilde{y})$, i.e., $L_t = \{\ell_1, \ldots, \ell_t\}$.

In this paper, we assume that the training losses in $L_t$ conform to a *Markov process*, which is to represent a changing system under the assumption that future states only depend on the current state (the Markov property) [43]. More specifically, at the $i$-th iteration, if we exploit an optimization algorithm for parameter updates (e.g., the stochastic gradient descent algorithm [4]) and omit other dependencies (e.g., $\tilde{S}$), we will have $P(\mathbf{w}_i | \mathbf{w}_{i-1}, \ldots, \mathbf{w}_0) = P(\mathbf{w}_i | \mathbf{w}_{i-1})$, which means that the future state of the classifier $f$ only depends on the current state. Furthermore, given a training example and the parameters of the classifier $f$, we can determine the loss of the training example as discussed. Therefore, the training losses in $L_t$ will also conform to a Markov process.

### 2.2 Extended Time Intervals

As limited time interval cannot address the instability issue of the estimation for the noisy class posterior well [42], we extend time intervals and exploit the training losses at different training iterations for sample selection. One straightforward idea is to use the *mean* of training losses at different training iterations. Hence, the selection criterion could be

$$\tilde{\mu} = \frac{1}{t} \sum_{i=1}^{t} \ell_i. \tag{1}$$

It is intuitive and reasonable to use such a selection criterion for sample selection, since the operation of averaging can mitigate the risks caused by the unstable estimation for the noisy class posterior, following better generalization. Nevertheless, such a method could arguably achieve suboptimal classification performance for learning with noisy labels. The main reason is that, due to the great harm of mislabeled data, part of training losses are with too large uncertainty and could be seen as outliers. Therefore, it could be biased to use the mean of training losses consisting of such outliers [10], which further influences sample selection. More evaluations for our claims are provided in Section 3.

### 2.3 Robust Mean Estimation and Conservative Search

We extend time intervals and meanwhile exploit the training losses at different training iterations more robustly. Specifically, we build two robust mean estimators from the perspectives of *soft truncation* and *hard truncation* [7]. Note that for specific tasks, it is feasible to decide the types of robust mean estimation with statistical tests based on some assumptions [8]. We leave the analysis as future work. Two *distribution-free* robust mean estimators are introduced as follows.

**Soft truncation.** We extend a classical M-estimator from [7] and exploit the *widest* possible choice of the *influence function*. More specifically, give a random variable $X$, let us consider a non-decreasing

influence function $\psi : \mathbb{R} \to \mathbb{R}$ such that

$$\psi(X) = \log(1 + X + X^2/2), X \geq 0. \tag{2}$$

The choice of $\psi$ is inspired by the *Taylor expansion of the exponential function*, which can make the estimation results more robust by reducing the side effect of extremum *holistically*. The illustration for this influence function is provided in Appendix A.1. For our task, given the observations on training losses, i.e., $L_t = \{\ell_1, \ldots, \ell_t\}$, we estimate the mean robustly as follows:

$$\tilde{\mu}_s = \frac{1}{t} \sum_{i=1}^{t} \psi(\ell_i). \tag{3}$$

We term the above robust mean estimator (3) the *soft estimator*.

**Hard truncation.** We propose a new robust mean estimator based on hard truncation. Specifically, given the observations on training losses $L_t$, we first exploit the K-nearest neighbor (KNN) algorithm [27] to remove some underlying outliers in $L_t$. The number of outliers is denoted by $t_o (t_o < t)$, which can be *adaptively determined* as discussed in [70]. Note that we can also employ other algorithms, e.g., principal component analysis [45] and the local outlier factor [6], to identify underlying outliers in $L_t$. The main reason we employ KNN is because of its relatively low computation costs [70].

The truncated loss observations on training losses are denoted by $L_{t-t_o}$. We then utilize $L_{t-t_o}$ for the mean estimation. As the potential outliers are removed with high probability, the robustness of the estimation results will be enhanced. We denote such an estimated mean as $\tilde{\mu}_h$. We have

$$\tilde{\mu}_h = \frac{1}{t - t_o} \sum_{\ell_i \in L_{t-t_o}} \ell_i. \tag{4}$$

The corresponding estimator (4) is termed the *hard estimator*.

We derive concentration inequalities for the soft and hard estimators respectively. The search strategy for less selected examples and overall selection criterion are then provided. Note that we do not need to explicitly quantify the mean of training losses. We only need to sort the training examples based on the proposed selection criterion and then use the selected examples for robust training.

**Theorem 1.** *Let $Z_n = \{z_1, \cdots, z_n\}$ be an observation set with mean $\mu_z$ and variance $\sigma^2$. By exploiting the non-decreasing influence function $\psi(z) = \log(1 + z + z^2/2)$. For any $\epsilon > 0$, we have*

$$\left| \frac{1}{n} \sum_{i=1}^{n} \psi(z_i) - \mu_z \right| \leq \frac{\sigma^2(n + \frac{\sigma^2 \log(\epsilon^{-1})}{n^2})}{n - \sigma^2}, \tag{5}$$

*with probability at least $1 - 2\epsilon$.*

Proof can be found in Appendix A.1.

**Theorem 2.** *Let $Z_n = \{z_1, \ldots, z_n\}$ be a (not necessarily time homogeneous) Markov chain with mean $\mu_z$, taking values in a Polish state space $\Lambda_1 \times \ldots \times \Lambda_n$, and with a minimal mixing time $\tau_{\min}$. The truncated set with hard truncation is denoted by $Z_{n_o}$, with $n_o < n$. If $|z_i|$ is upper bounded by $Z$. For any $\epsilon_1 > 0$ and $\epsilon_2 > 0$, we have*

$$\left| \frac{1}{n - n_o} \sum_{z_i \in Z_n \backslash Z_{n_o}} - \mu_z \right| \leq \frac{1}{n - n_o} \left( 2Z \sqrt{2\tau_{\min} \log \frac{2}{\epsilon_1}} + \frac{2Z n_o}{n} \sqrt{2\tau_{\min} \log \frac{2n}{\epsilon_2}} \right), \tag{6}$$

*with probability at least $1 - \epsilon_1 - \epsilon_2$.*

Proof can be found in Appendix A.2. For our task, let the training loss be upper-bounded by $L$. The value of $L$ can be determined easily by training networks on noisy datasets and observing the loss distribution [1].

**Conservative search and selection criteria.** In this paper, we will use the concentration inequalities (5) and (6) to present conservative search and the overall sample selection criterion. Specifically, we exploit their *lower bounds* and consider the selected number of examples during training. The selection of the examples that are less selected is encouraged.

---

**Algorithm 1** CNLCU Algorithm.

---

1: **Input** $\theta_1$ and $\theta_2$, learning rate $\eta$, fixed $\tau$, epoch $T_k$ and $T_{\max}$, iteration $t_{\max}$;

**for** $T = 1, 2, \ldots, T_{\max}$ **do**

   2: **Shuffle** training dataset $\tilde{S}$;

   **for** $t = 1, \ldots, t_{\max}$ **do**

      3: **Fetch** mini-batch $\bar{S}$ from $\tilde{S}$;

      4: **Obtain** $\bar{S}_1 = \arg\min_{S':|S'|\geq R(T)|\bar{S}|} \ell^\star(\theta_1, S')$;      // calculated with Eq. (7) or Eq. (8)

      5: **Obtain** $\bar{S}_2 = \arg\min_{S':|S'|\geq R(T)|\bar{S}|} \ell^\star(\theta_2, S')$;      // calculated with Eq. (7) or Eq. (8)

      6: **Update** $\theta_1 = \theta_1 - \eta\nabla\ell(\theta_1, \bar{S}_2)$;

      7: **Update** $\theta_2 = \theta_2 - \eta\nabla\ell(\theta_2, \bar{S}_1)$;

   **end**

   8: **Update** $R(T) = 1 - \min\left\{\frac{T}{T_k}\tau, \tau\right\}$;

**end**

9: **Output** $\theta_1$ and $\theta_2$.

---

Denote the number of times one example was selected by $n_t (n_t \leq t)$. Let $\epsilon = \frac{1}{2t}$. For the circumstance with soft truncation, the selection criterion is

$$\ell_s^\star = \tilde{\mu}_s - \frac{\sigma^2(t + \frac{\sigma^2 \log(2t)}{t^2})}{n_t - \sigma^2}. \tag{7}$$

Let $\epsilon_1 = \epsilon_2 = \frac{1}{2t}$, for the situation with hard truncation, by rewriting (6), the selection criterion is

$$\ell_h^\star = \tilde{\mu}_h - \frac{2\sqrt{2\tau_{\min}}L(t + \sqrt{2}t_o)}{(t - t_o)\sqrt{t}}\sqrt{\frac{\log(4t)}{n_t}}. \tag{8}$$

Note that we directly replace $t$ with $n_t$. If an example is rarely selected during training, $n_t$ will be far less than $n$, which causes the lower bounds to change drastically. Hence, we do not use the mean of all training losses, but use the mean of training losses in fixed-length time intervals. More details about this can be checked in Section 3.

For the selection criteria (7) and (8), we can see that they consist of two terms and have one term with a minus sign. The first term in Eq. (7) (or Eq. (8)) is to reduce the uncertainty of small-loss examples, where we use robust mean estimation on training losses. The second term, i.e., the statistical confidence bound, is to encourage the network to choose the less selected examples (with a small $n_t$). The two terms are constraining and balanced with $\sigma^2$ or $\tau_{\min}$. To avoid introducing strong assumptions on the underlying distribution of losses [8], we tune $\sigma$ and $\tau_{\min}$ with a noisy validation set. For the mislabeled data, although the model has high uncertainties on them (i.e., a small $n_t$) and tends to pick them, the overfitting to the mislabeled data is harmful. Also, the mislabeled data and clean data are rather hard to distinguish in some cases as discussed. Thus, we should search underlying clean data in a conservative way. In this paper, we initialize $\sigma$ and $\tau_{\min}$ with small values. This way can reduce the adverse effects of mislabeled data and meanwhile select the clean examples with large losses, which helps generalize. More evaluations will be presented in Section 3.

The overall procedure of the proposed method, which **c**ombats **n**oisy **l**abels by **c**oncerning **u**ncertainty (CNLCU), is provided in Algorithm 1. CNLCU works in a mini-batch manner since all deep learning training methods are based on stochastic gradient descent. Following [12], we exploit two networks with parameters $\theta_1$ and $\theta_2$ respectively to teach each other. Specifically, when a mini-batch $\bar{S}$ is formed (Step 3), we let two networks select a small proportion of examples in this mini-batch with Eq. (7) or (8) (Step 4 and Step 5). The number of instances is controlled by the function $R(T)$, and two networks only select $R(T)$ percentage of examples out of the mini-batch. The value of $R(T)$ should be larger at the beginning of training, and be smaller when the number of epochs goes large, which can make better use of memorization effects of deep networks [12] for sample selection. Then, the selected instances are fed into its peer network for parameter updates (Step 6 and Step 7).

## 3 Experiments

In this section, we evaluate the robustness of our proposed method to noisy labels with comprehensive experiments on the synthetic balanced noisy datasets (Section 3.1), synthetic imbalanced noisy datasets (Section 3.2), and real-world noisy dataset (Section 3.3).

## 3.1 Experiments on Synthetic Balanced Noisy Datasets

**Datasets.** We verify the effectiveness of our method on the manually corrupted version of the following datasets: *MNIST* [22], *F-MNIST* [58], *CIFAR-10* [21], and *CIFAR-100* [21], because these datasets are popularly used for the evaluation of learning with noisy labels in the literature [12, 65, 54, 23]. The four datasets are class-balanced. The important statistics of the used synthetic datasets are summarized in Appendix B.1.

**Generating noisy labels.** We consider broad types of label noise: (1). Symmetric noise (abbreviated as Sym.) [53, 31, 26]. (2) Asymmetric noise (abbreviated as Asym.) [32, 57, 52]. (3) Pairflip noise (abbreviated as Pair.) [12, 65, 71]. (4). Tridiagonal noise (abbreviated as Trid.) [68]. (5). Instance noise (abbreviated as Ins.) [9, 56]. The noise rate is set to 20% and 40% to ensure clean labels are diagonally dominant [32]. More details about above noise are provided in Appendix B.1. We leave out 10% of noisy training examples as a validation set.

**Baselines.** We compare the proposed method (Algorithm 1) with following methods which focus on sample selection, and implement all methods with default parameters by PyTorch, and conduct all the experiments on NVIDIA Titan Xp GPUs. (1). S2E [62], which properly controls the sample selection process so that deep networks can better benefit from the memorization effects. (2). MentorNet [16], which learns a curriculum to filter out noisy data. We use self-paced MentorNet in this paper. (3). Co-teaching [12], which trains two networks simultaneously and cross-updates parameters of peer networks. (4). SIGUA [13], which exploits stochastic integrated gradient underweighted ascent to handle noisy labels. We use self-teaching SIGUA in this paper. (5). JoCor [52], which reduces the diversity of networks to improve robustness. Other types of baselines such as *adding regularization* are provided in Appendix B.2. Note that we do not compare the proposed method with some state-of-the-art methods, e.g., SELF [39] and DivideMix [24]. It is because their proposed methods are aggregations of multiple techniques. We mainly focus on sample selectionin in learning with noisy labels. Therefore, the comparison is not fair. Here, we term our methods with soft truncation and hard truncation as CNLCU-S and CNLCU-H respectively.

**Network structure and optimizer.** For *MNIST*, *F-MNIST*, and *CIFAR-10*, we use a 9-layer CNN structure from [12]. Due to the limited space, the experimental details on *CIFAR-100* are provided in Appendix B.3. All network structures we used here are standard test beds for weakly-supervised learning. For all experiments, the Adam optimizer [20] (momentum=0.9) is used with an initial learning rate of 0.001, and the batch size is set to 128 and we run 200 epochs. We linearly decay learning rate to zero from 80 to 200 epochs as did in [12]. We take two networks with the same architecture but different initializations as two classifiers as did in [12, 65, 52], since even with the same network and optimization method, different initializations can lead to different local optimal [12]. The details of network structures can be checked in Appendix C.

For the hyper-parameters $\sigma^2$ and $\tau_{\min}$, we determine them in the range $\{10^{-1}, 10^{-2}, 10^{-3}, 10^{-4}\}$ with a noisy validation set. Here, we assume the noise level $\tau$ is known and set $R(T) = 1 - \min\{\frac{T}{T_k}\tau, \tau\}$ with $T_k$=10. If $\tau$ is not known in advanced, it can be inferred using validation sets [29, 66]. As for performance measurement, we use test accuracy, i.e., *test accuracy = (# of correct prediction) / (# of testing)*. All experiments are repeated five times. We report the mean and standard deviation of experimental results.

**Experimental results.** The experimental results about test accuracy are provided in Table 1, 2, and 3. Specifically, for *MNIST*, as can be seen, our proposed methods, i.e., CNLCU-S and CNLCU-H, produce the best results in the vast majority of cases. In some cases such as asymmetric noise, the baseline S2E outperforms ours, which benefits the accurate estimation for the number of selected small-loss examples. For *F-MNIST*, the training data becomes complicated. S2E cannot achieve the accurate estimation in such situation and thus has no great performance like it got on *MNIST*. Our methods achieve varying degrees of lead over baselines. For *CIFAR-10*, our methods once again outperforms all the baseline methods. Although some baseline, e.g., Co-teaching, can work well in some cases, experimental results show that it cannot handle various noise types. In contrast, the proposed methods achieve superior robustness against broad noise types. The results mean that our methods can be better applied to actual scenarios, where the noise is diversiform.

**Ablation study.** We first conduct the ablation study to analyze the sensitivity of the length of time intervals. In order to *avoid too dense figures*, we exploit *MNIST* and *F-MNIST* with the mentioned noise settings as representative examples. For CNLCU-S, the length of time intervals is chosen in

| Noise type | Sym. | | Asym. | | Pair. | | Trid. | | Ins. | |
|---|---|---|---|---|---|---|---|---|---|---|
| Method/Noise ratio | 20% | 40% | 20% | 40% | 20% | 40% | 20% | 40% | 20% | 40% |
| S2E | 98.46 | 95.62 | **99.05** | **98.45** | 98.56 | 94.22 | **99.02** | 97.23 | 97.93 | 94.02 |
| | ±0.06 | ±0.91 | ±0.02 | ±0.26 | ±0.32 | ±0.79 | ±0.09 | ±1.26 | ±1.26 | ±2.39 |
| MentorNet | 95.04 | 92.08 | 96.32 | 90.86 | 93.19 | 90.93 | 96.42 | 93.28 | 94.65 | 90.11 |
| | ±0.03 | ±0.42 | ±0.17 | ±0.97 | ±0.17 | ±1.54 | ±0.09 | ±1.37 | ±0.73 | ±1.26 |
| Co-teaching | 97.53 | 95.62 | 98.25 | 95.08 | 96.05 | 94.16 | 98.05 | 96.18 | 97.96 | 95.02 |
| | ±0.12 | ±0.30 | ±0.08 | ±0.43 | ±0.96 | ±1.37 | ±0.06 | ±0.85 | ±0.09 | ±0.39 |
| SIGUA | 92.31 | 91.88 | 93.96 | 62.59 | 93.77 | 86.22 | 94.92 | 83.46 | 92.90 | 86.34 |
| | ±1.10 | ±0.92 | ±0.82 | ±0.15 | ±1.40 | ±1.75 | ±0.83 | ±2.98 | ±1.82 | ±3.51 |
| JoCor | 98.42 | 98.04 | 98.05 | 94.55 | 98.01 | 96.85 | 98.45 | 96.98 | 98.62 | 96.07 |
| | ±0.14 | ±0.07 | ±0.37 | ±1.08 | ±0.19 | ±0.43 | ±0.17 | ±0.25 | ±0.06 | ±0.31 |
| CNLCU-S | **98.82** | **98.31** | 98.93 | 97.67 | **98.86** | **97.71** | **99.09** | **98.02** | **98.77** | **97.78** |
| | **±0.03** | **±0.05** | ±0.06 | ±0.22 | **±0.06** | **±0.64** | **±0.04** | **±0.17** | **±0.08** | **±0.25** |
| CNLCU-H | **98.70** | **98.24** | **99.01** | **98.01** | **98.44** | **97.37** | 98.89 | **97.92** | **98.74** | **97.42** |
| | **±0.06** | **±0.06** | **±0.04** | **±0.03** | **±0.19** | **±0.32** | ±0.15 | **±0.05** | **±0.16** | **±0.39** |

Table 1: Test accuracy (%) on *MNIST* over the last ten epochs. The best two results are in bold.

| Noise type | Sym. | | Asym. | | Pair. | | Trid. | | Ins. | |
|---|---|---|---|---|---|---|---|---|---|---|
| Method/Noise ratio | 20% | 40% | 20% | 40% | 20% | 40% | 20% | 40% | 20% | 40% |
| S2E | 89.99 | 75.32 | 89.00 | 81.03 | 88.66 | 67.09 | 89.53 | 77.29 | 88.65 | 79.35 |
| | ±2.07 | ±5.84 | ±0.95 | ±1.93 | ±1.32 | ±4.03 | ±2.63 | ±3.97 | ±2.12 | ±3.04 |
| MentorNet | 90.37 | 86.53 | 89.69 | 67.21 | 87.92 | 83.70 | 88.74 | 85.63 | 87.52 | 83.27 |
| | ±0.17 | ±0.65 | ±0.19 | ±2.94 | ±1.08 | ±0.49 | ±0.33 | ±0.59 | ±0.15 | ±1.42 |
| Co-teaching | 91.48 | 88.80 | 91.03 | 68.07 | 90.77 | 86.91 | 91.24 | 89.18 | 90.60 | 87.90 |
| | ±0.10 | ±0.29 | ±0.14 | ±4.58 | ±0.23 | ±0.71 | ±0.11 | ±0.36 | ±0.12 | ±0.45 |
| SIGUA | 87.64 | 87.23 | 76.97 | 45.96 | 69.59 | 68.93 | 79.97 | 76.14 | 76.92 | 74.89 |
| | ±1.29 | ±0.72 | ±2.59 | ±3.40 | ±5.75 | ±2.80 | ±3.23 | ±4.24 | ±5.09 | ±4.84 |
| JoCor | 91.97 | 89.96 | 90.95 | 79.79 | 91.52 | 87.40 | 92.01 | 89.42 | 91.43 | 87.59 |
| | ±0.13 | ±0.19 | ±0.21 | ±2.39 | ±0.24 | ±0.58 | ±0.17 | ±0.33 | ±0.71 | ±0.94 |
| CNLCU-S | **92.37** | **91.45** | **92.57** | **83.14** | **92.04** | **88.20** | 92.24 | 90.08 | **91.69** | **89.02** |
| | **±0.15** | **±0.28** | **±0.15** | **±1.77** | **±0.26** | **±0.44** | ±0.17 | ±0.34 | **±0.10** | **±1.02** |
| CNLCU-H | **92.42** | **91.60** | **92.60** | **82.69** | **91.70** | **87.70** | **92.33** | **90.22** | **91.50** | **88.79** |
| | **±0.21** | **±0.19** | **±0.18** | **±0.43** | **±0.18** | **±0.69** | **±0.26** | **±0.71** | **±0.21** | **±1.22** |

Table 2: Test accuracy on *F-MNIST* over the last ten epochs. The best two results are in bold.

the range from 3 to 8. For CNLCU-H, the length of time intervals is chosen in the range from 10 to 15. Note that the reason for their different lengths is that their different mechanisms. Specifically, CNLCU-S holistically changes the behavior of losses, but does not remove any loss from the loss set. We thus do not need too long length of time intervals. As a comparison, CNLCU-H needs to remove some outliers from the loss set as discussed. The length should be longer to guarantee the number of examples available for robust mean estimation. The experimental results are provided in Appendix B.4, which show the proposed CNLCU-S and CNLCU-H are robust to the choices of the length of time intervals. Such robustness to hyperparameters means our methods can be applied in practice and does not need too much effect to tune the hyperparameters.

Furthermore, since our methods concern uncertainty from two aspects, i.e., the uncertainty from both small-loss and large-loss examples, we conduct experiments to analyze each part of our methods. Also, as mentioned, we compare robust mean estimation with non-robust mean estimation when learning with noisy labels. More details are provided in Appendix B.4.

## 3.2 Experiments on Synthetic Imbalanced Noisy Datasets

**Experimental setup.** We exploit *MNIST* and *F-MNIST*. For these two datasets, we reduce the number of training examples along with the labels from "0" to "4" to 1% of previous numbers. We term such synthetic imbalanced noisy datasets as *IM-MNIST* and *IM-F-MNIST* respectively. This setting aims to simulate the extremely imbalanced circumstance, which is common in practice. Moreover, we exploit asymmetric noise, since these types of noise can produce more imbalanced case [41, 32]. Other settings such as the network structure and optimizer are the same as those in experiments on synthetic balanced noisy datasets.

| Noise type | Sym. | | Asym. | | Pair. | | Trid. | | Ins. | |
|---|---|---|---|---|---|---|---|---|---|---|
| Method/Noise ratio | 20% | 40% | 20% | 40% | 20% | 40% | 20% | 40% | 20% | 40% |
| S2E | 80.78 | 69.72 | 84.03 | 75.04 | 81.72 | 61.50 | 81.44 | 64.39 | 79.89 | 62.42 |
| | ±0.88 | ±3.94 | ±1.01 | ±1.24 | ±0.93 | ±4.63 | ±0.59 | ±2.82 | ±0.26 | ±3.11 |
| MentorNet | 80.92 | 74.67 | 80.37 | 71.69 | 77.98 | 69.39 | 78.02 | 71.56 | 77.02 | 68.17 |
| | ±0.48 | ±1.17 | ±0.26 | ±1.06 | ±0.31 | ±1.73 | ±0.29 | ±0.93 | ±0.71 | ±2.52 |
| Co-teaching | 82.35 | 77.96 | 83.87 | 73.43 | 80.94 | 72.81 | 81.17 | **74.37** | 79.92 | 73.29 |
| | ±0.16 | ±0.39 | ±0.24 | ±0.62 | ±0.46 | ±0.92 | ±0.60 | **±0.64** | ±0.57 | ±1.62 |
| SIGUA | 78.19 | 77.67 | 75.14 | 52.76 | 74.41 | 61.91 | 75.75 | 74.05 | 74.34 | 67.98 |
| | ±0.22 | ±0.41 | ±0.36 | ±0.68 | ±0.81 | ±5.27 | ±0.53 | ±0.41 | ±0.39 | ±1.34 |
| JoCor | 80.96 | 76.65 | 81.39 | 69.92 | 80.33 | 71.62 | 79.03 | 74.33 | 78.21 | 71.46 |
| | ±0.25 | ±0.43 | ±0.74 | ±1.63 | ±0.20 | ±1.05 | ±0.13 | ±1.09 | ±0.34 | ±1.27 |
| CNLCU-S | **83.03** | **78.25** | **85.06** | **75.34** | **83.16** | **73.19** | **82.77** | 74.37 | **82.03** | **73.67** |
| | **±0.21** | **±0.70** | **±0.17** | **±0.32** | **±0.25** | **±1.25** | **±0.32** | ±1.37 | **±0.37** | **±1.09** |
| CNLCU-H | **83.03** | **78.33** | **84.95** | **75.29** | **83.39** | **73.40** | **82.52** | **74.79** | **81.93** | **73.58** |
| | **±0.47** | **±0.50** | **±0.27** | **±0.80** | **±0.68** | **±1.53** | **±0.71** | **±1.13** | **±0.25** | **±1.39** |

Table 3: Test accuracy (%) on *CIFAR-10* over the last ten epochs. The best two results are in bold.

As for performance measurements, we use test accuracy. In addition, we exploit the selected ratio of training examples with the imbalanced classes, i.e., *selected ratio=(# of selected imbalanced labels / # of all selected labels)*. Intuitively, a higher selected ratio means the proposed method can make better use of training examples with the imbalanced classes, following better generalization [18].

**Experimental results.** The test accuracy achieved on *IM-MNIST* and *IM-F-MNIST* is presented in Figure 2. Recall the experimental results in Table 1 and 2, we can see that the imbalanced issue is *catastrophic* to the sample selection approach when learning with noisy labels. For *IM-MNIST*, as can be seen, all the baselines have serious overfitting in the early stages of training. The curves of test accuracy drop dramatically. As a comparison, the proposed CNLCU-S and CNLCU-H can give a try to large-loss but less selected data which are possible to be clean but equipped with imbalanced labels. Therefore, our methods always outperform baselines clearly. In the case of Asym. 10%, our methods achieve nearly 30% lead over baselines. For *IM-F-MNIST*, we can also see that our methods perform well and always achieve about 5% lead over all the baselines. Note that due to the huge challenge of this task, some baseline, e.g., S2E, has a large error bar. In addition, the baseline SIGUA performs badly. It is because SIGUA exploits stochastic integrated gradient underweighted ascent on large-loss examples, which makes the examples with imbalanced classes more difficult to be selected than them in other sample selection methods.

The selected ratio achieved on *IM-MNIST* and *IM-F-MNIST* is presented in Table 4. The results explain well why our methods perform better on synthetic imbalanced noisy datasets, i.e., our methods can make better use of training examples with the imbalanced classes. Note that since we give a try to large-loss but less selected data in a conservative way, the selected ratio is still far away from the class prior probability on the test set, i.e., 10%. However, a little improvement of the selection ratio can bring a considerable improvement of test accuracy. These results tell us that, in the sample selection approach when learning with noisy labels, improving the selected ratio of training examples with the imbalanced classes is challenging but promising for generalization. This practical problem deserves to be studied in depth.

### 3.3 Experiments on Real-world Noisy Datasets

**Experimental setup.** To verify the efficacy of our methods in the real-world scenario, we conduct experiments on the noisy dataset *Clothing1M* [59]. Specifically, for experiments on *Clothing1M*, we use the 1M images with noisy labels for training and 10k clean data for test respectively. Note that we do not use the 50k clean training data in all the experiments. For preprocessing, we resize the image to 256×256, crop the middle 224×224 as input, and perform normalization. The experiments on *Clothing1M* are performed once due to the huge computational cost. We leave 10% noisy training data as a validation set for model selection. Note that we do not exploit the resampling trick during training [24]. Here, *Best* denotes the test accuracy of the epoch where the validation accuracy was optimal. *Last* denotes test accuracy of the last epoch. For the experiments on *Clothing1M*, we use a ResNet-18 pretrained on ImageNet as did in [52]. We also use the Adam optimizer and set the batch size to 64. During the training stage, we run 15 epochs in total and set the learning rate $8 \times 10^{-4}$, $5 \times 10^{-4}$, and $5 \times 10^{-5}$ for 5 epochs each.

| Dataset | *IM-MNIST* | | | | *IM-F-MNIST* | | | |
|---|---|---|---|---|---|---|---|---|
| Method/Noise ratio | 10% | 20% | 30% | 40% | 10% | 20% | 30% | 40% |
| S2E | 0.13 | 0.11 | 0.09 | 0.05 | 0.13 | 0.17 | 0.16 | 0.12 |
| | ±0.12 | ±0.05 | ±0.02 | ±0.01 | ±0.04 | ±0.03 | ±0.02 | ±0.04 |
| MentorNet | 0.10 | 0.15 | 0.12 | 0.13 | 0.12 | 0.15 | 0.09 | 0.14 |
| | ±0.02 | ±0.02 | ±0.03 | ±0.02 | ±0.01 | ±0.03 | ±0.01 | ±0.02 |
| Co-teaching | 0.09 | 0.07 | 0.05 | 0.12 | 0.17 | 0.04 | 0.13 | 0.07 |
| | ±0.03 | ±0.02 | ±0.01 | ±0.01 | ±0.05 | ±0.00 | ±0.04 | ±0.01 |
| SIGUA | 0.04 | 0.04 | 0.01 | 0.02 | 0.03 | 0.02 | 0.04 | 0.00 |
| | ±0.00 | ±0.00 | ±0.00 | ±0.00 | ±0.00 | ±0.00 | ±0.00 | ±0.00 |
| JoCor | 0.11 | 0.08 | 0.07 | 0.06 | 0.05 | 0.13 | 0.13 | 0.07 |
| | ±0.04 | ±0.01 | ±0.03 | ±0.02 | ±0.01 | ±0.04 | ±0.03 | ±0.02 |
| CNLCU-S | **0.60** | **0.37** | **0.39** | **0.38** | **0.35** | **0.39** | **0.36** | **0.30** |
| | **±0.11** | **±0.09** | **±0.04** | **±0.06** | **±0.03** | **±0.04** | **±0.03** | **±0.02** |
| CNLCU-H | **0.57** | **0.32** | **0.37** | **0.32** | **0.34** | **0.35** | **0.32** | **0.28** |
| | **±0.13** | **±0.01** | **±0.07** | **±0.05** | **±0.02** | **±0.06** | **±0.04** | **±0.03** |

Table 4: Selected ratio (%) on *IM-MNIST* and *IM-F-MNIST*. The best two results are in bold.

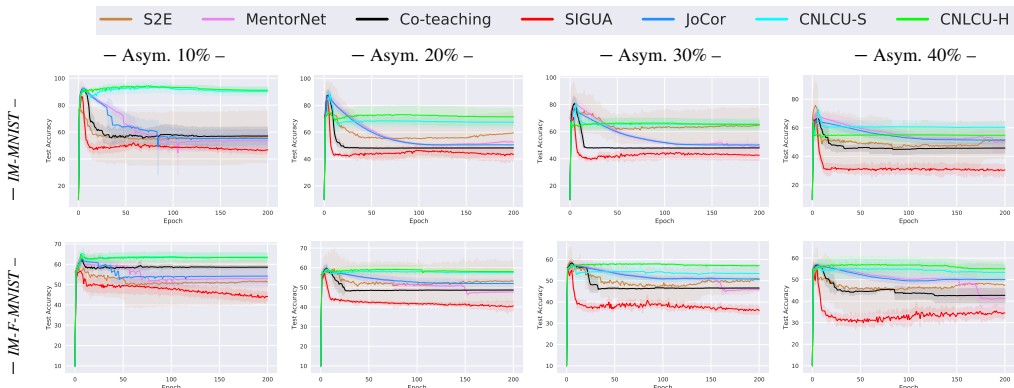

Figure 2: Test accuracy vs. number of epochs on *IM-MNIST* and *IM-F-MNIST*. The error bar for standard deviation in each figure has been shaded.

**Experimental results.** The results on *Clothing1M* are provided in Table 5. Specifically, the proposed methods get better results than state-of-the-art methods on *Best*, which achieve an improvement of +1.28% and +0.99% over the best baseline JoCor. Likewise, the proposed methods outperform all the baselines on *Last*. We achieve an improvement of +1.01% and +0.54% over JoCor. Note that the results are a bit lower than some state-of-art methods, e.g., [64] and [46], because of the following reasons. (1). We follow [52] and use ResNet-18 as a backbone. The state-of-art methods [64, 46] use ResNet-50 as a backbone. Our aim is to make the experimental results directly comparable with previous papers [52] in the same area. (2). We only focus on the sample selection approach and do not employ other advanced techniques, e.g., introducing the prior distribution [46] and combining semi-supervised learning [24, 39, 28].

| Methods | S2E | MentorNet | Co-teaching | SIGUA | JoCor | CNLCU-S | CNLCU-H |
|---|---|---|---|---|---|---|---|
| *Best* | 67.34 | 68.36 | 69.37 | 62.89 | 70.09 | **71.37** | **71.08** |
| *Last* | 65.90 | 67.42 | 68.62 | 58.73 | 69.75 | **70.76** | **70.29** |

Table 5: Test accuracy (%) on *Clothing1M*. The best two results are in bold.

# 4 Conclusion

In this paper, we focus on promoting the prior sample selection in learning with noisy labels, which starts from concerning the uncertainty of losses during training. We robustly use the training losses at different iterations to reduce the uncertainty of small-loss examples, and adopt confidence interval estimation to reduce the uncertainty of large-loss examples. Experiments are conducted on benchmark datasets, demonstrating the effectiveness of our method. We believe that this paper opens up new possibilities in the topics of using sample selection to handle noisy labels, especially in improving the robustness of models on imbalanced noisy datasets.

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
