# OpenReview forum: "Sample Selection with Uncertainty of Losses for Learning with Noisy Labels"
_NeurIPS.cc/2021/Conference — NeurIPS 2021 Submitted_

### Official Review · Reviewer_1zBg · 2021-07-14

**Rating:** 6
**Confidence:** 4

**Summary:**

This paper proposes a CNLCU strategy that incorporates the uncertainty of losses by adopting interval estimation.  In learning with noisy labels, the sample selection approach regards small-loss data as correctly labeled during training.  This paper gives large-loss data a try to explore underrepresented data that are correctly labeled but seem to be mislabeled at first glance. Experiments demonstrate that the proposed method is superior to baselines and robust to a broad range of label noise types. This paper opens up new possibilities in the topics of using sample selection to handle noisy labels.

**Limitations And Societal Impact:**

Yes

**Main Review:**

In learning with noisy labels, the sample selection based on small-loss criteria is the most common method. This paper reveals the limitation of small-loss criteria and proposes a new selection criterion based on uncertainty to give large-loss data a try. The theoretical analyses are detailed and the claims are well supported by experimental results.  This paper adopts interval estimation instead of point estimation, where the lower bounds are used for sample selection. In this paper, the writer focuses on promoting the prior sample selection. However, this paper is not well organized and there is quite a bit of room for improvement in the representation. The overall writing is not fluent and the theorems are lack of interpretation. Besides, the penultimate paragraph in the introduction is a little bit confusing and figure 2 is worthy of improvement.

**Time Spent Reviewing:**

4

---

> ### Author Response · Authors · 2021-08-05
> **The Response to Reviewer 1zBg**
>
> Thank you for the valuable feedback! We carefully answer your questions one by one.
>
> **Q1:** The improvement for the writing and more interpretation for theoretical analyses.
> **A1:** We will take your advice.
> - Some words and sentences of this paper may be improper, which makes readers confused. We will check them carefully to improve the writing.
> - Compared with theoretical results, the interpretation and explanation for theoretical analyses are not enough. The intuition for them is needed, which will help readers understand this paper. We will add them as suggested in our ﬁnal version and ﬁnal supplementary material.
>
> **Q2:** The concerns on “Introduction” and Figure 2.
> **A2:** We address your concerns and take your advice in the following.
> - The penultimate paragraph in the introduction is about the justification of the proposed methods philosophically and technically. We think that our presentation may make you confused. A more detailed and smooth explanation for it will be added in our revision version to enhance this paper.
> - For Figure 2, we think that it may be a bit dense, which influences its readability. Also, some discussions are needed to add. We will improve it and add analyses and discussions for it.

---

### Official Review · Reviewer_PRnD · 2021-07-15

**Rating:** 7
**Confidence:** 5

**Summary:**

This paper introduces the uncertainty of losses into the sample selection approach for learning with noisy labels. The authors discuss how to effectively exploit underlying clean examples that are less selected during training. Experimental results verify the effectiveness of the proposed method.

**Limitations And Societal Impact:**

The proposed method introduces extra hyperparameters into training, which cannot be determined adaptively. Although the results look very promising, it is better if such issues can be addressed.

The authors should discuss the negative societal impacts of this work like the possible model bias from the data.

**Main Review:**

The sample selection approach is popular and useful in learning with noisy labels. Motivated by the fact that deep networks first memorize the training data with clean labels, the training data with small losses can be seen as clean data and used for optimization. This paper considers the uncertainty of losses in the sample selection procedure, which consists of two aspects: the uncertainty of small-loss data and the uncertainty of large-loss data. The authors use the concentration inequalities and confidence bounds to study these uncertainties at the same time. The experiments in this paper are comprehensive. The results are also promising, especially when the training data are noisy and imbalanced.

Strength
- Strong motivation. This paper is motivated by experimental results, which show that sometimes the loss values cannot fully reflect the purity of an example. When the data are noisy and imbalanced, the losses of mislabeled data and clean imbalanced data are almost equal and hence cannot be used directly for sample selection. Besides, the loss achieved at a certain epoch may not be trusted. The results are clear and well motivate this paper.
- Great writing. The logic of this paper is smooth. The authors first reveal the issues of prior effects and then present the proposed method step by step. Most of the technical details are described clearly.
- Theory. This paper provides theoretical analyses for two robust mean estimators, which could guide the process of sample selection during training.
- Experiments. The experiments of this paper are comprehensive. The authors consider five types of synthetic label noise and real-world label noise, following good experimental results. Detailed experimental analyses and ablation study are also provided.

Weakness/Question
- Learning with noisy imbalanced data is challenging. Expect for the results (Figure 1 Right) on the MNIST dataset, would the results on the other datasets be similar to the case on MNIST?
- The methods CNLCU-S (H) are built on Co-teaching. It seems that the proposed methods can also build on other sample selection methods. Could the authors further add such discussions and results?
- The proposed method relies on a noisy validation set to find suitable hyperparameters, i.e., $\sigma$ and $\tau$, which makes the methods more complex.
- The authors use ResNet-18 for the experiments on Clothing1M. I agree with the claims of the authors. The proposed method cannot be competitive with DivideMix. However, most of the existing methods use a ResNet-50 network. It would be better if the results with ResNet-50 can be provided.
- The paper discusses the weaknesses of existing methods on sample selection at a high level (Section 1). Could the authors provide a more detailed algorithm flow like Algorithm 1 to explain this more clearly?

It is expected that the above issues/concerns could be addressed.



**Time Spent Reviewing:**

5

---

> ### Author Response · Authors · 2021-08-06
> **The Response to Reviewer PRnD**
>
> Thanks for your review and suggestions! We will answer the questions one by one.
>
> **Q1:** Would the results on the other datasets be similar to the case on IM-MNIST?
> **A1:** Yes. We exploit the results of IM-MNIST for our motivation. The results on the other imbalanced datasets are similar to the case on IM-MNIST, i.e., the clean but imbalanced data have large losses, which makes prior sample selection methods not work well. In Section 3.2, the experiments on IM-F-MNIST also confirm our claims.
>
> **Q2:** The experimental results when the proposed methods are built on other sample selection methods.
> **A2:** We study the uncertainty of losses which are built on MentorNet and S2E. Let us describe the experiments in detail.
> - The experiments are conducted on MNIST, CIFAR-10, and IM-MNIST. The noise rate is set to 40%.
> - We use “MentorNet-S(H)” (resp. S2E-S(H)) to denote that MentorNet (resp. S2E) is boosted with our methods.
> - The experimental results are shown in Table 2-1 and 2-2. With the study of the uncertainty of losses, the methods based on the small-losses trick can be enhanced clearly, especially on IM-MNIST, which supports our claims well.
>
> | Methods | Sym. 40\% (MNIST) | Asym. 40% (MNIST) | Pair. 40% (MNIST) | Trid. 40% (MNIST) | Ins. 40% (MNIST) | Asym. 40% (IM-MNIST)|
> | :----: | :----: | :----: | :----: | :----: | :----: | :----: |
> | MentorNet | 92.08$\pm$0.42 | 90.86$\pm$0.97 | 90.93$\pm$1.54 | 93.28$\pm$1.37 | 90.11$\pm$1.26 | 50.39$\pm$7.84 |
> | MentorNet-S |  **93.52$\pm$0.41** | **91.33$\pm$0.50** | **92.06$\pm$0.67** | **94.06$\pm$0.66** | **91.35$\pm$0.90** | **60.12$\pm$3.06**|
> | MentorNet-H | **93.07$\pm$0.66** | **90.95$\pm$0.84** | **92.08$\pm$0.78** | **93.29$\pm$1.04** | **92.03$\pm$0.92** | **61.52$\pm$4.18** |
> | S2E | 95.62$\pm$0.91 | 98.45$\pm$0.26 | 94.22$\pm$0.79 | 97.23$\pm$1.26 | 94.02$\pm$2.39 | 51.42$\pm$13.04 |
> | S2E-S | **95.73$\pm$0.85** | **98.76$\pm$0.33** | **94.72$\pm$0.90** | **97.64$\pm$0.29** | **94.73$\pm$0.92**| **59.77$\pm$6.34**|
> | S2E-H | **95.77$\pm$0.62** | **98.66$\pm$0.29** | **94.83$\pm$0.63** | **97.64$\pm$0.58** | **94.69$\pm$1.70**| **58.72$\pm$6.06**|
>
> Table 2-1: The test accuracy (%) on noisy MNIST and IM-MNIST.
>
> | Methods | Sym. 40\% | Asym. 40%| Pair. 40% | Trid. 40% | Ins. 40% |
> | :----: | :----: | :----: | :----: | :----: | :----: |
> | MentorNet | 74.67$\pm$1.17 | 71.69$\pm$1.06 | 69.39$\pm$1.73 | 71.56$\pm$0.93 | 68.17$\pm$2.52 |
> | MentorNet-S | **75.61$\pm$0.93** | **72.03$\pm$0.93** | **69.89$\pm$1.50** | **72.33$\pm$0.55** | **69.33$\pm$1.29** |
> | MentorNet-H | **74.88$\pm$1.00** | **71.87$\pm$1.06** | **70.09$\pm$1.31** | **71.77$\pm$1.06** | **68.57$\pm$2.19** |
> | S2E | 69.72$\pm$3.94 | 75.04$\pm$1.24 | 61.50$\pm$4.63 | 64.39$\pm$2.82 | 62.42$\pm$3.11 |
> | S2E-S | **72.72$\pm$2.06** | **75.22$\pm$0.93** | **65.29$\pm$2.73** | **66.78$\pm$1.37** | **66.77$\pm$1.30** |
> | S2E-H | **72.84$\pm$1.58** | **75.17$\pm$0.88** | **65.43$\pm$2.04** | **67.69$\pm$2.31** | **65.82$\pm$2.95** |
>
> Table 2-2: The test accuracy (%) on noisy CIFAR-10.
>
> **Q3:** The concerns about the hyperparameters of the proposed methods.
> **A3:** We justify the claims of using hyperparameters.
> - Although the hyperparameters make the methods more complex, the use of hyperparameters aims to reduce the dependency on strong assumptions and thus make our methods perform well in practice.
> - The experiments on real-world label noise datasets could verify the effectiveness of our methods.
> - We will discuss the strong assumptions which are used for estimating $\sigma$ and $\tau$ and their limitations in our final paper.
>
> **Q4:** The experiments on Clothing1M with a pre-trained ResNet-50 network structure.
> **A4:** We conduct the experiments on Clothing1M to further justify our claims. The pre-trained ResNet-50 network structure is used as suggested. The experimental results are provided in Table 2-3. As can be seen, our methods outperform baselines clearly.
>
> Note that we also take the advice of another reviewer and combine other advanced methods to boost our method as did in DivideMix. The experiments are also conducted with ResNet-50.
>
> | Methods | S2E | MentorNet | Co-teaching | SIGUA | JoCor | CNLCU-S | CNLCU-H|
> | :----: | :----: | :----: | :----: | :----: | :----: | :----: | :----: |
> | Best | 68.03 | 67.25 | 67.94 | 65.37 | 69.06 | **71.57** | **71.22** |
> | Last | 66.25 | 66.59 | 67.05 | 60.77 | 68.41 | **70.88** | **70.46** |
>
> Table 2-3: The test accuracy (%) on Clothing1M with a pre-trained ResNet-50 network structure.
>
> **Q5:** A more detailed algorithm flow like Algorithm 1 is needed.
> **A5:** We will add an algorithm flow to discuss the weaknesses of the prior sample selection method in detail.

---

> > ### Comment · Reviewer_PRnD · 2021-08-12
> > **Recommend Acceptance**
> >
> > The authors have addressed my concerns well. Especially the experiments on other sample selection methods are helpful in understanding the generalization of the proposed method.
> >
> > Overall, I like the paper and increase evaluation to accept.

---

### Official Review · Reviewer_mZp6 · 2021-07-15

**Rating:** 6
**Confidence:** 4

**Summary:**

The authors discuss potential weakness of previous sample selection criteria and propose new selection criteria based on the uncertainty of losses (not losses themselves).  Some theoretical analysis on the criteria is provided.  Then, they experimentally validate the proposed algorithms (CNLCU-S and CNLCU-H), showing that the proposed algorithms outperform some baselines in learning with noisy labels.

**Limitations And Societal Impact:**

I think that there is no potential negative societal impact of this work.

**Main Review:**

Distinguishing between mislabeled examples and hard examples is a fundamentally difficult problem and worthwhile to investigate.

The key idea is to incorporate the uncertainty of losses by adopting interval estimation instead of point estimation of losses.  This idea is definitely reasonable, but is not totally new.  Although the details are different, batch selection based on uncertainty has been actively discussed in this field.  For example, refer to ActiveBias (NeurIPS 2017).

The assumption that the loss conforms to a Markov process is not reasonable.  Thus, Theorem 2 based on this assumption may not work well with a real loss function.

My main concern is on the experiments.  The authors argue that they did not compare the proposed algorithms with the state-of-the-art algorithms (e.g., SELF and DivideMix), because they focus on sample selection mechanisms.  I understand the authors' intention.  However, I believe that this is not acceptable for this top-tier conference, because the proposed algorithms should be proven to improve the cutting edge performance.  Sample selection alone may not be very convincing, because combining various techniques is a clear trend these days.

According to a nice survey, https://arxiv.org/abs/2007.08199, there are recent sample selection methods, which are not covered in this paper.  Please see Section III.E.  Even though the authors want to focus on sample selection, it would be necessary to cover recent advances more comprehensively.

Also, it is common to include at least two real-world noisy datasets, e.g., WebVision, Clothing1M.

Typo:
Line 217, selectionin

===================

After the author feedback: the authors have successfully addressed my concern on the experiments (lack of the comparison with the state-of-the-art algorithms). Thus, I am willing to raise my rating to 6.

**Time Spent Reviewing:**

5

---

> ### Author Response · Authors · 2021-08-05
> **The Response to Reviewer mZp6 (Part 1)**
>
> Thank you for your constructive comments and suggestions! We will answer all the questions.
>
> **Q1:** More discussions on related work ActiveBias (NeurIPS 2017) are needed.
> **A1:** Thank you for mentioning this interesting and solid paper, which exploits lightweight estimates of sample uncertainty to improve generalization. In general, the differences between our work and ActiveBias lie in the different problems, different focuses, and different technical implementation. In more detail, we summarize the main differences between the two works.
> - ActiveBias mainly focuses on mining hard examples to help generalization. Although it mentions the label-noise problem, it does not consider comprehensive noise settings. Also, the baselines are SGD, ADAM, and other optimization methods, but not the methods are specially designed for learning with noisy labels. In contrast, our work mainly focuses on learning with noisy labels and has more comprehensive experiments, which explore the performance of different label-noise methods under noisy supervision.
> - ActiveBias focuses on using the hard examples, but not too difficult examples, which is stated in the original paper. In the imbalanced setting of this paper, clean imbalanced data are too difficult to distinguish with the predictions of models (Figure 1 right) and therefore are too difficult examples. Our methods aim to make use of such examples. Experiments show that such examples are of great importance for generalization.
> - The introduction of uncertainty in ActiveBias relies on the estimation of the prediction variance but does not consider the bad influence of mislabeled data/outliers on this estimation. Our methods use the loss distribution and the side effect of mislabeled data is considered by robust mean estimators.
>
> **Q2:** The reasonability of the assumption that the loss conforms to a Markov process.
> **A2:** We justify the reasonability of the assumption that the loss conforms to a Markov process.
> - The assumption for Theorem 2 is inspired by the modern optimization method for deep neural networks, e.g., SGD and Mini Batch SGD, where the weights of a network in the next iteration only depend on the weights in the current iteration.
> - Although some optimization methods, e.g., AdaDelta and Adam, exploit the momentum which uses the gradients in multiple iterations to stabilize or speed up training, the weight updates still mainly rely on the weights in the last iteration.
> - This assumption is built on the training loss but does not rely on a specific loss function.
> - A series of experimental results, especially for the results on real-world label-noise datasets, justify the reasonability of this assumption.
>
> **Q3 & Q5:** The main concerns on more experiments for comparing with the state-of-the-art algorithms. In addition, more experiments on real-world label-noise datasets are needed.
> **A3 & A5:** We agree with your opinion and have taken your valuable advice. We compare our methods with the SOTA method, i.e., DivideMix [1], which has released codes (https://github.com/LiJunnan1992/DivideMix). Let us describe the experiments in detail.
> - We conduct experiments on three real-world datasets, i.e., Food-101, Webvision (mini), and Clothing1M, which are popularly used in learning with noisy labels.
> - We use the reported performance in [1] for Webvision (mini) and Clothing1M. We test the performance of DivideMix on Food-101 by using a pre-trained ResNet-50 model.
> - We follow the paradigm of DivideMix and replace its GMM model with our methods for sample selection. Other settings stay the same.
> - The results are provided in Table 1-1. DivideMix-S (resp. DivideMix-H) means that our CNLCU-S (resp. CNLCU-H) is combined with the advanced techniques in DivideMix.
> - By combining the advanced techniques as did in DivideMix, the robustness of models can be further improved. In some cases, the improvement may be not very significant. This is because: (1) It is harder to gain significant improvement on challenging real-world datasets (see [2]), especially in the comparison with the strong baseline DivideMix; (2) We do not tune the combination of advanced techniques finely and only use the proposed sample selection paradigm with them.
> - Consistent improvements in multiple cases show that the proposed algorithms could improve the cutting edge performance.
>
>
> | Methods / Datasets | Food-101 | Webvision (mini) | Clothing1M |
> | :----: | :----: | :----: | :----: |
> | DivideMix | 86.73 | 77.32 | 74.76 |
> | DivideMix-S | **86.92** | **77.53** | **74.90** |
> | DivideMix-H | **86.88** | **77.48** | **74.82** |
>
> Table 1-1: The test accuracy (%) on three real-world datasets.
>
> ----
> [1] Junnan Li et al. Dividemix: learning with noisy labels as semi-supervised learning. ICLR, 2020.
> [2] Pengfei Chen et al. Understanding and utilizing deep neural networks trained with noisy labels. ICML 2019.

---

> > ### Author Response · Authors · 2021-08-05
> > **The Response to Reviewer mZp6 (Part 2)**
> >
> > **Q4:** More baselines on sample selection for learning with noisy labels are needed.
> > **A4:** We have checked this paper carefully, where the sample selection methods can be divided into three categories: (a) “Multi-network Learning”, (b) “Multi-round Learning”, and (c) “Hybrid Approach”. We add more baselines for comparison. Let us describe the experiments in detail.
> > - As our method belongs to (a), we exploit INCV [2] which belongs to (b), and NPCL [3] which belongs to (c), to make the comparison more comprehensive.
> > - We conduct experiments on MNIST, CIFAR-10, and IM-MNIST respectively. The noise rate is set to 40%. Other settings stay the same as our paper.
> > - The detailed results are provided in Tables 1-2 and 1-3. As can be seen, in most cases, our methods can achieve the best accuracy. For IM-MNIST, our methods outperform NPCL and INCV clearly. In the other cases, e.g., Sym. 40%, Pair. 40% on MNIST and Sym. 40% on CIFAR-10, our methods could be competitive with these two baselines.
> >
> > Following your constructive suggestions and recommended survey, we will consider more types of baselines on sample selection in the final version.
> >
> > | Methods | Sym. 40\% (MNIST) | Asym. 40% (MNIST) | Pair. 40% (MNIST) | Trid. 40% (MNIST) | Ins. 40% (MNIST) | Asym. 40% (IM-MNIST)|
> > | :----: | :----: | :----: | :----: | :----: | :----: | :----: |
> > | NPCL | 98.21$\pm$0.11| 96.14$\pm$1.21 | **97.50$\pm$0.18** | 97.62$\pm$0.29 | 95.75$\pm$1.04 | 50.29$\pm$4.98 |
> > | INCV | **98.37$\pm$0.06** | 97.67$\pm$0.26 | 97.21$\pm$0.39 | 97.65$\pm$0.32 | 94.97$\pm$1.35 | 48.65$\pm$6.32 |
> > | CNLCU-S | **98.31$\pm$0.05** | **97.67$\pm$0.22** | **97.71$\pm$0.64** | **98.02$\pm$0.17** | **97.78$\pm$0.25** | **60.34$\pm$3.93** |
> > | CNLCU-H | 98.24$\pm$0.06 | **98.01$\pm$0.03** | 97.37$\pm$0.32 | **97.92$\pm$0.05** | **97.42$\pm$0.39**| **54.71$\pm$5.13** |
> >
> > Table 1-2: The test accuracy (%) on noisy MNIST and IM-MNIST.
> >
> > | Methods | Sym. 40\% | Asym. 40%| Pair. 40% | Trid. 40% | Ins. 40% |
> > | :----: | :----: | :----: | :----: | :----: | :----: |
> > | NPCL | **78.62$\pm$0.32** | 74.50$\pm$0.73 | 71.35$\pm$2.73 | 74.26$\pm$1.65 | 71.93$\pm$1.65 |
> > | INCV | 77.93$\pm$0.67 | 73.08$\pm$1.55 | 72.65$\pm$2.02 | 74.15$\pm$1.05 | 70.77$\pm$2.66 |
> > | CNLCU-S | 78.25$\pm$0.70 | **75.34$\pm$0.32** | **73.19$\pm$1.25** | **74.37$\pm$1.37** | **73.67$\pm$1.09**|
> > | CNLCU-H | **78.33$\pm$0.50** | **75.29$\pm$0.80** | **73.40$\pm$1.53** | **74.79$\pm$1.13** | **73.58$\pm$1.39** |
> >
> > Table 1-3: The test accuracy (%) on noisy CIFAR-10.
> >
> > **Q6:** The typo in Line 217.
> > **A6:** Thanks. “selectionin” -> “selection”. We will correct it in the revision paper.
> >
> > ----
> > [3] Yueming Lyu and Ivor W. Tsang. Curriculum Loss: Robust Learning and Generalization against Label Corruption. ICLR 2020.

---

### Author Response · Authors · 2021-08-08
**The Response to Reviewers**

Dear reviewers:

Thanks a lot for your efforts in reviewing this paper. We tried our best to address all mentioned concerns/problems. Are there unclear explanations here? We could further clarify them.

Best,
Authors

---

### Decision · Program_Chairs · 2021-09-27

**Decision:**

Reject

**Comment:**

The submission considers the problem of learning from noisy labels, and propose to incorporate an interval estimate of training loss to the sample selection approach. The intuition is to distinguish between mislabelled data and under-represented data. Theoretical and empirical results shows the promise of the method.

Three reviewers have carefully considered the strengths and weaknesses of the paper. During the author rebuttal period, the authors put in significant effort to address some concerns about empirical results raised by the reviewers. The reviewers appreciated the clarifications, and one reviewer increased their score. There was unfortunately no substantive discussion among the reviewers post rebuttal.

There are many high quality papers submitted to NeurIPS each year, and this year is no exception. Many papers are on the borderline, with reviewers finding the problems important, but also found weaknesses in the paper. Unfortunately this paper could not be accepted into this year's program.